# The Potential Impact of Different Taxation Scenarios towards Sugar-Sweetened Beverages on Overweight and Obesity in Brazil: A Modeling Study

**DOI:** 10.3390/nu14235163

**Published:** 2022-12-04

**Authors:** Carla Cristina Enes, Ana Elisa M. Rinaldi, Luciana Bertoldi Nucci, Alexander Itria

**Affiliations:** 1Center for Life Sciences, Postgraduate Program in Health Sciences, Pontifical Catholic University of Campinas (PUC-Campinas), Avenue John Boyd Dunlop, s/n, Campinas 13060-904, Brazil; 2Postgraduate Program in Health Sciences, School of Medicine, Federal University of Uberlândia (UFU), 1720, Pará Avenue, Block 2U, Uberlândia 38400-902, Brazil; 3Management and Technology Sciences Center, University of Sao Carlos (UFSCar), João Leme dos Santos Highway, SP-264, Km 110-Itinga, Sorocaba-SP, Sorocaba 18052-780, Brazil

**Keywords:** sugar-sweetened beverages, fiscal policy, obesity, body mass index, tax

## Abstract

The adoption of fiscal policies based on the specific taxation of sugar-sweetened beverages (SSBs) has been recommended by international health agencies, as they are measures that potentially reduce consumption. This study is an ex ante risk comparison that estimates the impact of three tax scenarios (20, 25, and 30%) with a 100% pass-on rate to SSBs on the prevalence of high weight and obesity in the Brazilian population. Data on the consumption habits, weight, and height of 46,164 adults aged 20 years or over from Brazilian recent national surveys were used. The shift in consumption after taxation was estimated based on the price elasticity of the demand. The percentage changes in overweight for 20, 25, and 30% taxation were 1.84% (95%CI: 1.82; 1.86), 1.89% (95%CI: 1.87; 1.90), and 2.25% (95%CI: 2.24; 2.27), respectively. The change in the prevalence of obesity was 1.93% (95%CI: 1.87; 2.00), 2.90% (95%CI: 2.80; 3.02), and 4.16% (95%CI: 4.01; 4.32), respectively. Taxes on SSBs may have a more favorable result among the heaviest consumers, who are young adults (20–29 years), especially men, thereby promoting a greater reduction in the prevalence of high weight and obesity.

## 1. Introduction

In recent decades, the prevalence of obesity has increased to a worrying degree, becoming the leading global and public health nutritional problem [1]. In Brazil, 60% of the adult population (approximately 95 million people) is overweight, and the prevalence of obesity increased from 12.2 to 26.8% between 2002/2003 and 2019 [2]. This scenario draws attention chiefly because obesity is associated with several chronic diseases, such as cardiovascular diseases, type 2 diabetes mellitus, and cardiometabolic diseases, which are the main causes of death and disability in Brazil [3]. Faced with the increase in obesity prevalence, Brazil was the first country to formally commit to the United Nations Decade of Action in Nutrition (2016–2025) in 2017, which offers a ten-year window of opportunity to intensify policies, programming and actions to improve nutrition. The Nutrition Decade should lead to the transformation of food systems in order to achieve global nutrition targets, the elimination of all forms of malnutrition, and accelerate the achievement of the 2030 Agenda. At the time, several ambitious goals were set, such as preventing the advance of obesity in the adult population and reducing the consumption of SSBs in at least 30% among adults until the year 2019 [4].

The World Health Organization highlights that diet is among the four main risk factors for the occurrence of chronic non-communicable diseases. The relationship between the consumption of SSBs, that is, drinks with added sugar, including non-diet soft drinks/sodas, flavored juice drinks, sports drinks, sweetened tea, coffee drinks, energy drinks, and electrolyte replacement drinks, and heightened risk of obesity has been evidenced in several studies [5,6,7], possibly due to the high sugar content present in these beverages. In addition, the intake of calories in a liquid form promotes incomplete compensation in total calories consumed in subsequent meals [8]. Data available from the most recent Family Budget Survey (acronym in Portuguese “*Pesquisa de Orçamentos Familiares*” POF 2017–2018) show that the Brazilian adult population consumes 61 L of SSBs a year on average, and about 65% of this volume is contained in soft drinks [9]. Although the average annual consumption of SSBs in Brazil is lower than in other Latin American countries such as Mexico (163 L/year of soft drinks) and Argentina (120–130 L/year of SSB) [10,11], in the past 20 years Brazilians have tripled their consumption of sugar from soft drinks [12]. Currently, calories from SSBs represent 2.1% of total calories consumed among the adult population [9]. 

It is estimated that the annually the consumption of SSBs in Brazil is responsible for the morbidity of 1,810,832 adults and 12,748 deaths [13]. Health spending is approximately USD 24.7 million a year for the care of issues related to weight and obesity associated with the consumption of SSBs. These costs amount to USD 527 million in the treatment of diabetes, renal and cardiovascular disorders, musculoskeletal disorders, cancer, and other diseases associated with the consumption of SSBs [14].

The adoption of fiscal policies based on the specific taxation of SSBs has been recommended by national and international health agencies, as they are measures that can potentially reduce consumption of this group of beverages. Evidence from modeling studies, systematic reviews, and meta-analyses conducted in the United Kingdom, Ireland, South Africa, Germany, Thailand, and Colombia, respectively, show that the taxation of these beverages may reduce their consumption, leading to a reduction in body weight [15,16,17,18,19,20,21,22,23].

Successful experiences in countries where taxation has already been implemented have been observed in Mexico; after the implementation of a tax in 2014 [10], the purchase of SSBs fell by 6% in the first year, increasing in the second year, reaching 10% [24]. France and a number of regions in the United States have identified a reduction in the consumption of SSBs and an increase in revenue after the implementation of taxes [25,26,27].

Although Brazil is a potential candidate for taxing SSBs, especially due to widespread consumption and the growing prevalence of obesity in recent decades, discussion about this fiscal policy remains incipient in the country. Between 2016 and 2019, ten bills for taxation of sweetened beverages were presented. Six of them aimed to raise taxes, and are currently under consideration in the Committees by the first legislative House. Four bills aimed at repealing an act which increased taxes on soft drink concentrates; these have been rejected and archived [28]. In 2020, another action to promote healthy eating was approved in the form of a new nutritional labeling model front for processed and ultra-processed food packaging to indicate when there is an excess of critical nutrients [29]. Recently, in May 2022, a bill called ‘CIDE soft drinks’ (Contribution for Intervention in the Economic Domain) (or ‘*CIDE refrigerantes*’ acronym in Portuguese)” was approved in the federal senate, which provides for a 20% increase in taxes on SSB. The bill that created CIDE levies a tax on the commercialization and import of soft drinks and sugary drinks. The proposal determines the application of funds collected in public health actions and services and sports projects. Currently, “CIDE soft drinks” is under evaluation by the economic affairs commission, with no expectation of approval.

Simulations on the effect of tax measures on the demand for SSBs in Brazil have been presented; nevertheless, until now, no study has estimated the impact of taxation on prevalence of overweight and obesity [30,31,32]. Thus, the aim of this study was to estimate the impact of different tax scenarios for SSBs on the prevalence of overweight and obesity in the Brazilian population. 

## 2. Methods

### 2.1. Data and Assumptions

This is an ex ante risk comparative study. We estimated the impact of taxation on all SSBs as provided for in the CIDE soft drinks bill. Considering that soft drinks represent 65% of all SSBs consumed by the Brazilian population, we simulated the impact of taxing only sugar-sweetened soft drinks on the outcomes of interest (Appendix A). To calculate the impact of three tax scenarios on SSBs in reducing overweight and obesity, we used the estimated price elasticities (own- and cross-price elasticities) of demand for SSB. We considered cross-price elasticities in order to capture change in beverage consumption behavior in response to the change in taxed SSB prices. Non-taxed beverages considered in the model in order to evaluate the substitution effect were fruit juice, coffee and tea, milk, water. 

From own- and cross-price elasticity, the change in the consumption of all beverages imposed by taxation as well as the change in energy intake were estimated in order to model the effect on body weight, and consequent prevalence of overweight and obesity. The estimate of the effect of taxation on body weight was based on the theoretical model already been used in other countries such as South Africa, Ireland, Thailand, and the United Kingdom [15,16,19,21]. The model underlying this study (Figure 1) assumed that an SSB tax affects consumption of different beverages via elasticity of demand, which in turn affects energy consumption and, consequently, body weight and BMI. The references for different data sources are additionally shown.

### 2.2. Price Elasticity and Pass-on Rate

Own-price elasticity can be interpreted as the percentage change that occurs in the demand of a product when there is a one percentage point change in price, while cross-price elasticity is the percentage change in purchases of a good when the price of another good changes by one percent. The price elasticities employed here come from a study that used the most recent Household Budget Survey (POF 2017–2018) as a database [32], a nationally representative household survey, which investigated the structure of income, expenditure, and food and beverage consumption in households throughout Brazil. Cross- and own-price elasticities were estimated according to beverage categories (Appendix A).

The present study evaluates the impact of a hypothetical tax following CIDE on what is taxed, and assumes a pass-through rate of 100%. Three tax scenarios were simulated: scenario 1–20%; scenario 2–25%; and scenario 3–30%, as there is reasonable and growing evidence that properly planned taxes on SSBs would result in proportionate reductions in consumption, especially if aimed at increasing the retail price by at least 20% [33].

### 2.3. Data Sources, Sampling, and Consumption of SSBs in Brazil

Information on the consumption of SSBs used in this study comes from the most recent POF, 2017–2018. The analyzed data were obtained through the Brazilian Institute of Geography and Statistics (*Instituto Brasileiro de Geografia e Estatística*—IBGE) between July 11, 2017, and July 9, 2018 from a probabilistic sample of 57,920 households representative of the Brazilian population. The POF 2017–2018 was a cross-sectional survey that used a complex sampling plan by conglomerates, with the drawing of census tracts in a first stage and households in a second. The selection of primary sampling units (PSUs) to compose the master sample occurred independently in each geographic and socioeconomic stratum proportionally to the number of households in the sectoral grid of the 2010 Demographic Census. More details on the sampling process are available elsewhere [9]. Information on personal food consumption was obtained from a subsample of households (20,112 households and 46,164 individuals) randomly selected from the total survey sample. In this study, individuals aged 20 years or over were included. We did not include children in the analysis because consumption data were not available for this age group.

Food consumption was estimated with two 24-h recalls, including all foods consumed inside and outside the home. In this study, only the first recall was considered. All categories of SSBs and non-sweetened beverages were considered in the simulation. Mean calorie intake from beverages was estimated separately for each category (soft drinks, milk-sweetened beverages, sports and energy drinks, other sugary drinks, milk, fruit juice, and tea and coffee), both at baseline and across the three taxation scenarios for the total sample, according to sex and age groups.

### 2.4. Prevalence of Overweight and Obesity in Brazil

Self-reported weight and height data were obtained from the National Health Survey (*Pesquisa Nacional de Saúde*—PNS) conducted in 2019. From these data, the body mass index (weight in kilograms divided by height in meters squared—BMI) was calculated to estimate the number of adults and elderly people with excess weight and obesity. The prevalence of overweight (adults and elderly with BMI ≥ 25 kg/m^2^ and <30kg/m^2^) and obesity (adults and elderly with BMI ≥ 30 kg/m^2^) was estimated from the mean BMI for men and women aged 20 years or over. This was a population-based household survey with a cross-sectional design representative of the adult population of Brazil residing in private households in its territory. The study used complex sampling with a three-stage conglomerate plan. Information from individuals aged 20 years or over (*n* = 85,825) was used. Baseline data on BMI and overweight and obesity prevalence for adult population are presented in Appendix A.

### 2.5. Modeling

The model was used to compare a scenario without taxation (the baseline) and with taxation. The main parameters used in the model were the tax pass-on rate, cross- and own-price elasticity, beverage consumption, and body mass index. Three tax scenarios (20, 25, and 30%) with a pass-on rate of 100% were used to simulate the price increase in combination with the estimated price elasticity of demand for each beverage category in order to estimate the variation in consumption. From this simulation, the variation in daily energy intake was obtained. All parameters used in the estimation model are presented in Appendix A.

### 2.6. Changes in Calorie Consumption

The same percentage change in consumption was applied to both sexes and all age groups. Differences in consumption of beverages at baseline (before taxation) generated different absolute estimates of daily calorie intake by sex and age groups after taxation. It was assumed that the reduction in calories from the taxation of SSBs represented an equal reduction in total daily calorie intake. Compensatory changes in energy intake from other non-taxed beverage sources were considered. Energy expenditure from physical activity was not included in the model [34].

### 2.7. Changes in Body Weight and BMI

To estimate the changes in weight and BMI resulting from the reduction in calorie intake, the dynamic energy balance equation proposed by Hall et al. (2011) [35], which has been frequently used to estimate the potential impact of taxing SSBs on weight, was used [18,21,36,37]. The equation assumes that a reduction of 100 kJ per day leads to a reduction of one kilogram of body weight, with half the weight change being achieved in about one year and 95% of the weight change being achieved in about three years. Considering that the height remains the same, the result is a change in the BMI value. In our study, we applied a contrafactual which assumed only two different stationary states (a baseline population without tax as the reference and a population exposed to tax), assuming that the weight change would reach 95% at the end of three years post-tax. In addition, it was assumed that the same level of individual physical activity was maintained. The parameter of interest evaluated was BMI. Calculations were performed for the three taxation scenarios.

### 2.8. Change in Overweight and Obesity Prevalence

To estimate the result of the different taxation scenarios on the prevalence of overweight and obesity, only individuals classified as overweight or obese at baseline (before taxation) were considered, which was compared to the prevalence in the three taxation scenarios.

### 2.9. Sensitivity Analyses

Theoretically, and assuming an ideal competitive market condition, the tax pass-through rate to the final consumer should be 100%, as simulated in our base model. However, because many markets, including the beverage market, do not have perfect competition, the tax can be partially absorbed by the manufacturer in the production process [38]. For this reason, we simulated a model in which the tax pass-through rate was 80% in a sensitivity analysis. 

## 3. Results

Analyzing individual consumption data at baseline from the POF 2017–18, the mean consumption of SSBs in the Brazilian population is approximately 176 (95%CI: 169, 184) mL/person/day, which is equivalent to a mean intake of 67 (95%CI: 64, 70) kcal/person/day (Appendix A). Among the SSBs, sugar-sweetened soft drinks were the most consumed, representing approximately 65% of all SSBs, followed by the group of other SSBs (approximately 20%). It was observed that mean consumption varies according to age and sex. Younger adults (aged 20–29 years) are the greatest consumers of SSBs (268.6 mL/day), while those aged 70 years and over consume the smallest amounts (105.5 mL/day). Regarding gender, men were the main consumers of SSBs (208 mL versus 153 mL for women). The mean consumption of non-sweetened beverages (except for water) is approximately 576 mL/person/day, which is equivalent to a mean intake of 153 kcal/person/day (Appendix A). Coffee and tea and fruit juice were the most consumed categories, together representing approximately 94% of the total volume from caloric beverages. 

### 3.1. Changes in SSB Consumption and Energy Intake

Table 1 presents the mean change in energy intake after SSB taxation in the three simulated scenarios according to sex and age group. Daily calorie reduction was observed only for young adults (20–29 years); this was observed for both sexes, and was greater with the increase in the tax.

An increase in daily calorie intake was observed among individuals aged 30 years and over, ranging on average from 8 kcal for 20% tax to 12 kcal daily for 30% tax. The impact of taxation was greater among men compared to among women.

### 3.2. Changes in BMI

Table 2 shows the mean change in BMI considering the three tax scenarios. The estimated reduction in mean BMI was observed only among young adults (20–29 years), and was greater in men compared to women. Mean BMI variation was positive for men and women over the age of 30 years. In the scenario of 20% taxation of SSBs, BMI increased on average by 0.131 kg/m^2^ (95% CI: 0.130, 0.133), reaching 0.197 kg/m^2^ (95% CI: 0.195, 0.199) in the 30% taxation scenario.

### 3.3. Change in Overweight and Obesity Prevalence

Figure 2 shows the percentage changes in the prevalence of overweight and obesity in the three tax scenarios according to sex and age groups. Estimates were calculated for the population aged 20 years and over, and results refer to changes three years post-tax. The percentage change for overweight men with taxation of 20, 25, and 30% was 1.14% (95% CI: 1.08; 1.22), 1.14% (95% CI: 1.07; 1.21), and 1.70% (95% CI: 1.68; 1.72), respectively. For overweight women, the percentage change was 2.60% (95% CI: 2.59; 2.60), 2.71% (95% CI: 2.69; 2.72), and 2.86% (95% CI: 2.85; 2.87) for taxation of 20%, 25%, and 30%, respectively. This variation is equivalent to an increase of 881.601 (95% CI: 872.507, 890.695), 905.540 (95% CI: 893.059, 918.020), and 1.080.724 (95% CI: 1.062.865, 1.098.582) overweight for the three tax rates applied. 

There was a positive variation in the prevalence of obesity for men of 1.28% (95% CI: 1.23; 1.34), 2.06% (95% CI: 1.97; 2.16), and 3.62% (95% CI: 3.47; 3.79), respectively in the three taxation scenarios. Among women, the positive variation was 2.42% (95% CI: 2.35; 2.50), 3.54% (95% CI: 3.43; 3.66), and 4.56 (95% CI: 4.41; 4.73) for a 20%, 25%, and 30% tax, respectively. These results represent an increase of 639.342 (95% CI: 634.980;643.703) individuals with obesity for a 20% tax, 960.207 (95% CI: 956.293;964.120) for a 25% tax, and 1.370.825 (95% CI:1.370.558;1.379.091) for a 30% tax.

The percentage negative variation in overweight prevalence occurred only for young adults (20–29 years) for both men and women, and the negative variation of obesity prevalence was observed only among men aged 20–29 years. 

When analyzing the simulation of taxation only on sugar-sweetened soft drinks, the results were more encouraging. The percentage change for overweight men with taxation of 20, 25, and 30% was −0.72% (95% CI: −0.77, −0.67), −0.70% (95% CI: −0.75, −0.64), and −0.92% (95% CI: −0.98, −0.87), respectively. There was a negative variation in the prevalence of obesity for men of −2.11% (95% CI: −2.22, −2.01), −2.53% (95% CI: −2.67, −2.42), and −2.70% (95% CI: −2.84, −2.57), respectively, in the three taxation scenarios (Appendix A).

For overweight women, no percentage variation was observed in taxation of 20% (0.0%; 95% CI: −0.03, 0.02), while for 25 and 30% taxation the variation was positive at 0.39% (95% CI: 0.33, 0.45) and 0.59% (95% CI: 0.53, 0.64), respectively. There was a positive variation in the prevalence of obesity for women of 0.43% (95% CI: 0.42, 0.44), 0.78% (95% CI: 0.75, 0.80), and 0.71% (95% CI: 0.69, 0.73), respectively, in the three scenarios of taxation.

### 3.4. Sensitivity Analyses

Overall, the model with a lower pass-on rate (80%) yielded a lower degree of change in SSBs consumption, energy intake, and BMI in all populations. Overall, mean BMI with a pass-through rate of 80% for taxation of 20%, 25%, and 30% showed an increase of 0.113, 0.140, and 0.169 kg/m^2^, respectively, slightly lower values compared to the reduction of 0.131, 0.164, and 0.197 kg/m^2^ when considering the 100% pass-through rate. Percentage variations in overweight prevalence were higher in the three taxation scenarios analyzed, and were slightly lower for obesity. When evaluating the different taxes on SSBs, it was observed that the higher the value of the tax, the greater the positive percentage variation in the prevalence of overweight and obesity (Appendix A).

## 4. Discussion

The results of our study suggest that the creation of a tax on SSBs only has an impact in terms of reducing the prevalence of overweight and obesity among younger adults (20–29 years), in particular for men, which tends to be greater with the increase in the tax. Among younger women (20–29 years), the tax on SSBs reduces the prevalence of overweight only. The three scenarios analyzed, with taxes of 20, 25, and 30%, revealed that the expected variation in overweight three years post-tax would be 1.84%, 1.89%, and 2.25%, respectively, for the general population. The positive variation in the prevalence of obesity for 20, 25, and 30% tax was 1.93%, 2.90%, and 4.16%, respectively. For individuals aged 30 years or more, an increase in the prevalence of overweight and obesity was observed three years post-tax. This result can be explained by the increase in mean daily calorie intake among individuals 30 years or over due to the substitution effect of other beverages in response to taxation of SSBs. That is, in addition to the consumption of SSBs being reduced with advancing age, the consumption of non-sweetened beverages with a higher energy density than SSBs, such as milk, tends to increase among the elderly, leading to a positive variation in calorie intake and consequently in BMI.

By considering the cross-price elasticities in the present study, it was possible to estimate the scenario of consumption of other substitute beverages for SSBs in the face of taxation. The consumption of non-taxed beverages was higher with advancing age, especially milk, and due to their greater volume and calories we noticed an increase in daily caloric intake from these beverages after taxation. We emphasize, however, that based on the analysis performed here it was only possible to analyze the total balance of calories before and after taxation, without considering the nutritional quality of the beverage replacement. Although there is a positive balance in caloric value, and consequently in BMI and nutritional status among individuals with 30 years or over, taxation may lead to the replacement of beverages the main ingredients of which are sugars and chemical additives with drinks that have greater nutritional value, such as fruit juice and milk. We identified as another positive point of taxation the possibility of preventing obesity and other chronic diseases in early adulthood, as the greatest impacts were verified in adults under 30 years of age. Additionally, in a study performed with data from the Surveillance System for Risk and Protective Factors for Chronic Diseases by Telephone Survey (VIGITEL), it was identified that the persistence of obesity in early adulthood when it was diagnosed in late adolescence was 65% for men and 47% for women [39].

The prevalence of obesity showed a greater percentage variation of 1.93%, 2.90%, and 4.16%, respectively. In general, a tax on SSBs would mainly benefit men and younger adults under the age of 30 years, who are the heaviest consumers, corroborating the findings of other studies [15,16,20,21,37]. Although data from the most recent national survey show that the prevalence of overweight and obesity is higher among women [2], the results of this study are positive in that they reveal a greater impact of the taxation of SSBs among men, as their consumption is higher than that of women. The greatest impact of taxation on young adults may be positive, as this fiscal policy could play a preventive role in weight gain and the development of chronic diseases related to excessive consumption of sweetened beverages, reducing the prevalence and incidence of these diseases and consequently reducing health costs.

The positive impact of creating a tax for SSBs on the prevalence of obesity has already been evidenced in other modeling studies [15,18,19,20,21,37,40], which found estimates of obesity reduction to different degrees in the general population when simulating a 20% tax; differently, our findings that show a positive impact only among younger adults (20–29 years). Manyema et al. identified an estimated reduction of 3.8% in obesity prevalence for men and 2.4% for women in South Africa [19]. In Germany, Schwendicke and Stolpe found a reduction of 4% in obesity prevalence and 3% in overweight [20]. In India, an estimated reduction of 3% was found for overweight and obesity prevalence [18]. A recently published systematic review showed that taxation of SSBs had a positive effect on decreasing the prevalence of overweight and obesity in most studies, and that this impact was greater with higher taxation [23]. The same study showed that non-significant effects on sales, purchases, consumption, and obesity prevalence were observed in high-income countries, showing the differences in this impact between countries with different income classifications.

Variations in the percentage of overweight and obesity reduction observed between studies may be mainly attributed to differences in the consumption of SSBs and the prevalence of overweight and obesity at baseline. In addition, the parameters used in the models, such as own-price elasticity, inclusion of cross-price elasticity, the pass-on rates of consumer taxes, and differences in the types of beverages included in the group and in the energy-balance models all contribute to variations in results.

The positive mean variation in BMI identified in our study was contrary to other studies that have simulated a 20% tax [21,37], as well as the estimated increase in the prevalence of overweight and obesity. It is important to note that both studies mentioned included a greater variety of sweetened beverages and did not consider cross-elasticities in the estimates, and thus did not consider the effects of substitution by other beverages. In this case, estimates of the impact of taxation of SSBs tend to be overestimated.

The mean reduction in calorie intake was highly variable between studies due to the variation in the consumption of SSBs at baseline in different populations, directly impacting obesity reduction estimates. Countries such as Mexico, Thailand, the United Kingdom, the United States, and Germany present a mean consumption of SSBs higher than that used in this study [15,20,21,37,41]. The set of SSBs considered in the analyses may impact mean calorie intake as well. Thus, the inclusion of alcoholic beverages or only sweetened soft drinks and juices in the SSB group can potentially affect baseline calorie intake [15,16,19].

More than fifty territories around the world have already implemented different types of taxation on SSBs adapted to the economic, political, and cultural characteristics of each region [42]. This type of extra-fiscal taxation has as its main objective the change of behavior in relation to the consumption of a product more than the generation of revenue.

As well as modeling studies, observational studies that have evaluated the impact of taxing SSBs after implementation in countries such as Mexico, Chile, Barbados, France, and the United States have shown successful results, with a reduction in the purchase, sale, and consumption of this group of beverages and an increase in government revenue [25,26,42,43,44,45]. Contrary to what has been happening in several countries around the world, in Brazil there is currently a stimulus to the entire production chain of sugar-sweetened non-alcoholic beverages through tax incentives, which generated BRL 3.8 billion (Brazilian Real) annually in tax waivers in 2016 [46].

Obesity should not be understood as an individual problem; rather, it is a result of the dominance of production chains that prioritize the manufacture of ultra-processed beverages and foods. Marketing strategies practiced by transnationals in the beverage market influence the population’s choices, increasing the availability of these products to the detriment of healthier alternatives [47].

The complex Brazilian tax system highlights the need to institute a tax that is not easily absorbed by production costs in order to ensure that there is an impact on the final price. In addition, large transnationals in the SSBs sector could migrate their factories to other locations, as taxes levied on these beverages vary by state according to location. Thus, a federal tax along the lines of CIDE, which allows the linking of resources obtained to specific funds, programs, and actions to guarantee benefits to the population, could compensate for negative externalities by increasing investments in health. This strategy generates greater political and societal support for the implementation of fiscal measures and serves as a counterpoint to the regressive argument, as the most affected population benefits the most [48].

A recent study estimated that with the creation of a 20% tax on SSBs the government’s annual revenue would increase by BRL 4.7 billion, reflecting an increase of BRL 2.4 billion in the gross domestic product [32]. In addition, it is estimated that more than 69,000 new jobs can be created as a result of taxation, mainly due to the replacement of these beverages by other products, such as milk or natural juices, that demand more intensive labor [32].

However, in addition to taxation, other regulatory measures that bring positive impacts to the health and quality of life of the population are needed, such as the regulation of food advertising [49], adoption of nutritional labeling standards [50,51], in particular the inclusion of frontal warnings to help identify critical food nutrients such as sugars and fats, and the promotion of healthier environments [52], along with the prohibition of the sale of harmful products in institutional environments, especially in schools. Facilitating access to healthier foods through subsidies for the cultivation of fresh products, as well as increasing the supply of these foods in different regions of the city, are important actions that can ensure access to healthy and adequate food.

### Study Strengths and Limitations

Our results provide important contributions to the literature, as this study is one of the few to assess the impact of taxation of SSBs on the health of the population in low- and middle-income countries, and the first in Brazil to assess the impact on overweight and obesity, considering three different taxation scenarios. Several strengths of the present study can be highlighted. First, nationally representative data from recent surveys were used to estimate the consumption of SSBs inside and outside the home and the prevalence of overweight and obesity at baseline, thus increasing the generalizability of the results. Another strength of this study is that the cross- and own-price elasticity values used were estimated specifically for the Brazilian population by the Institute of Economic Research Foundation (*Fundação Instituto de Pesquisas Econômicas*—FIPE) [32], an organization responsible for creating important official economic indicators for the country based on recent information obtained in a national survey, the same one we used in this study. It is noteworthy that price elasticities were estimated for a wide group of SSBs by beverage category, as this indicator is known to vary between the types of beverages. To estimate the variation in body weight, a dynamic and conservative equation was used [35], reducing the chance of overestimating the variation in the prevalence of overweight and obesity. Finally, our model considered the substitution effect of non-sweetened beverages (milk, fruit juice, and coffee and tea) in response to SSB price increase, thereby avoiding overestimation of the daily calorie reduction and consequently of the variation in the prevalence of overweight and obesity.

Several limitations of the study need to be mentioned as well. First, there was no differentiation of price elasticity values according to sex, age group, or income. It is known that younger individuals are the heaviest consumers of SSBs, and evidence reveals that such consumers are less sensitive to price variations [34], which may suggest a lower response to taxation than estimated in this study for this population group. We did not incorporate the uncertainty of price elasticities because values were not available, which could result in a greater amplitude for the calculated CIs. 

This study did not consider the results of taxation by income group, and previous studies have shown that individuals with lower incomes consume more SSBs compared to those with higher incomes [53,54]. Studies suggest that there is a greater reduction in the demand for SSBs among low-income families [24,55], probably because the proportional expenditure on food for rich families is much lower compared to poor families, meaning that the variation in demand against the price increase is different.

The weight and height data used in the baseline are self-reported and underestimate the prevalence of obesity, leading to possible underestimation of the benefits of a tax. Finally, in estimating weight variation, the adopted equation assumes a linear association between calorie intake and body weight, leading to a new steady state after weight reduction. This assumption can be simplified given that different equations can be applied for men and women and that the basal metabolic rate can change with weight loss.

## 5. Conclusions

The results of this study suggest that the creation of a tax on SSBs has an impact on reducing the prevalence of overweight and obesity only among younger adults (20–29 years) and for men, which tends to be greater with the increase in the tax. Among women, the positive impact of taxation occurs only among younger women (20–29 years), reducing overweight prevalence only. 

We emphasize the importance of future studies to investigate the impact of taxation of SSBs on other health outcomes considering different socioeconomic levels. Studies that assess the cost-effectiveness of a fiscal policy for SSBs in Brazil are needed. Nonetheless, our results provide important estimates of the impact of an SSBs tax on the prevalence of obesity, which is an important risk factor for NCDs and the main causes of mortality worldwide. The implementation of policies to reduce SSB consumption may yield the additional advantage of providing a new source of public funds to support healthy lifestyles.

## Figures and Tables

**Figure 1 nutrients-14-05163-f001:**
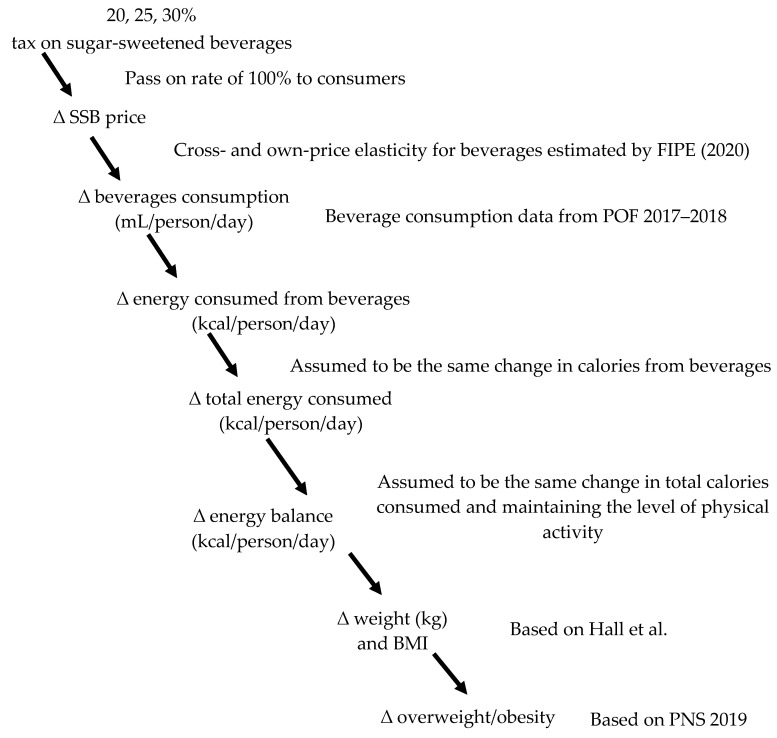
Modeled causal pathway between SSB taxation and overweight/obesity.

**Figure 2 nutrients-14-05163-f002:**
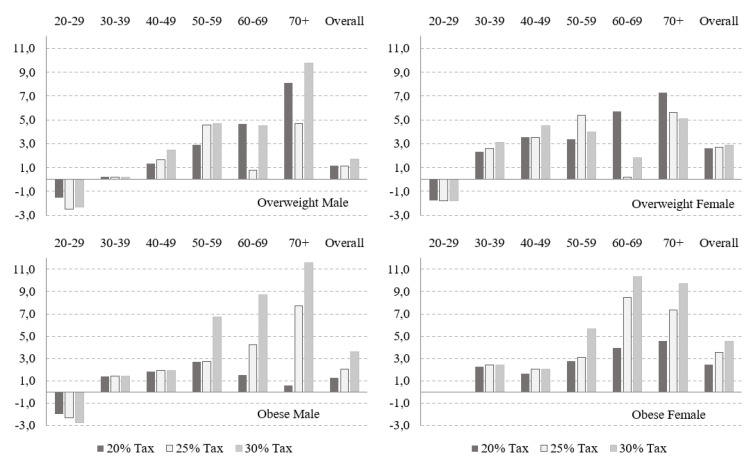
Percentage change in overweight and obesity prevalence (relative change in percent) by age group and sex as a result of a 20, 25, and 30% SSB tax.

**Table 1 nutrients-14-05163-t001:** Estimated mean change in calorie intake (kcal/person/day) in the three taxation scenarios according to sex and age group.

Sex	Age Group (Years)	20% Tax Mean (95% CI)	25% Tax Mean (95% CI)	30% Tax Mean (95% CI)
Male	20 to 29	−4.0 (−6.9; −1.2)	−5.1 (−8.6; −1.5)	−6.1 (−10.3; −1.8)
	30 to 39	3.0 (0.5; 5.5)	3.8 (0.6; 6.9)	4.5 (0.8; 8.3)
	40 to 49	9.0 (7.1; 10.9)	11.2 (8.8; 13.7)	13.5 (10.6; 16.4)
	50 to 59	12.1 (10.2; 10.9)	15.1 (12.8; 17.5)	18.2 (15.3; 21.0)
	60 to 69	13.2 (11.3; 15.2)	16.5 (14.1; 18.9)	19.8 (16.9; 22.7)
	≥70	14.8 (11.5; 18.1)	18.5 (14.4; 22.6)	22.2 (17.2; 22.7)
	All men	6.4 (5.2; 7.5)	7.9 (6.6; 9.3)	9.5 (7.9; 11.2)
Female	20 to 29	−0.1 (−2.3; 2.0)	−0.1 (−2.8; 2.6)	−0.2 (−3.4; 3.1)
	30 to 39	7.9 (6.3; 9.5)	9.8 (7.8; 11.9)	11.8 (9.4; 14.2)
	40 to 49	10.4 (9.0; 11.9)	13.1 (11.2; 14.9)	15.7 (13.5; 17.8)
	50 to 59	11.7 (10.0; 13.5)	14.7 (12.5; 16.8)	17.6 (15.0; 20.2)
	60 to 69	15.5 (13.2; 17.8)	19.3 (16.4; 22.2)	23.2 (19.7; 26.6)
	≥70	18.0 (16.1; 20.0)	22.5 (20.0; 25.0)	27.1 (24.1; 30.0)
	All women	9.5 (8.6; 10.3)	11.8 (10.8; 12.9)	14.2 (13.0; 15.4)
Overall		8.0 (7.2; 8.8)	10.0 (9.0; 11.0)	12.0 (10.8; 13.2)

**Table 2 nutrients-14-05163-t002:** Estimated mean change in BMI (kg/m^2^) in the three taxation scenarios according to sex and age group.

Sex	Age Groups	Mean Change in BMI in kg/m^2^ (95% CI)		
		Tax 20%	Tax 25%	Tax 30%
Male	20–29	−0.056 (−0.056; −0.055)	−0.071 (−0.071; −0.071)	−0.085 (−0.085; −0.084)
	30–39	0.042 (0.042; 0.042)	0.053 (0.053; 0.053)	0.063 (0.062; 0.063)
	40–49	0.128 (0.127; 0.128)	0.159 (0.158; 0.159)	0.192 (0.191; 0.192)
	50–59	0.174 (0.173; 0.174)	0.217 (0.216; 0.217)	0.261 (0.260; 0.262)
	60–69	0.193 (0.192; 0.194)	0.241 (0.240; 0.242)	0.290 (0.289; 0.291)
	≥70	0.219 (0.219; 0.220)	0.274 (0.273; 0.275)	0.329 (0.328; 0.330)
	All men	0.096 (0.094; 0.098)	0.119 (0.117; 0.122)	0.144 (0.141; 1.146)
Female	20–29	−0.002 (−0.002; −0.002)	−0.002 (−0.002; −0.002)	−0.003 (−0.003; −0.003)
	30–39	0.128 (0.128; 0.128)	0.159 (0.158; 0.159)	0.191 (0.191; 0.192)
	40–49	0.170 (0.170; 0.171)	0.214 (0.214; 0.215)	0.257 (0.256; 0.258)
	50–59	0.193 (0.193; 0.194)	0.243 (0.242; 0.244)	0.291 (0.290; 0.292)
	60–69	0.260 (0.259; 0.261)	0.324 (0.323; 0.325)	0.389 (0.388; 0.391)
	≥70	0.309 (0.308; 0.311)	0.387 (0.385; 0.388)	0.466 (0.464; 0.468)
	All women	0.163 (0.162; 0.165)	0.204 (0.202; 0.206)	0.245 (0.243; 0.247)
Overall		0.131 (0.130; 0.133)	0.164 (0.162; 0.166)	0.197 (0.195; 0.199)

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
