# Peer review of "The Potential Impact of Different Taxation Scenarios towards Sugar-Sweetened Beverages on Overweight and Obesity in Brazil: A Modeling Study"

_nutrients, 2022, doi:10.3390/nu14235163_

Round 1

Reviewer 1 Report

In the Abstract the authors use the abbreviation SSBs, without explanation in the first line. They give it on line 3.

In introduction, about the sentence: "Health spending is approximately 553.5 million USD a year for the care of diseases associated to the consumption of SSBs. These costs could reach 9.83 billion USD in 2050 if no action is taken [14]." Where did the authors find these values? because reference 14 makes expenditure estimates, with assumptions relating to the reduction of the BMI.

The third point of Figure 1 is not complete.

Introducing the formula for calculating the BMI in paragraph 2.7 is absurd: either it is inserted as soon as the BMI is mentioned or it is not inserted.

The same for the BMI ranges that identify obesity and overweight. They should be placed in paragraph 2.4, not paragraph 2.8.

In the discussion the authors write: "the urgency of actions such as the implementation of taxes on products that harm health is clear". One can easily agree with this statement. But the data and results presented in this study seem to say otherwise. In this study sugary drinks, unquestionably harmful to health, if taxed higher lead to an increase in weight problems. Thus the action of raising taxes appears not only useless, but also harmful. Perhaps a purchase containment policy could be more useful?

Clearly, the fact that aspects such as people's economic availability or level of education have not been considered is extremely limiting.

The fact that self-reported weight and height data may underestimate rates of overweight and obesity does not change the study's findings. The trends seem to be evident: higher taxation does not have a positive effect on weight.

Reducing the consumption of sugary drinks in the population that consumes the most can undoubtedly have a positive impact on weight as well. And this seems to be the only relevant data obtained from the study.

The other data are in contrast with the conclusions: what is the point of increasing taxation if this leads to a general increase in the BMI of people aged > 30 years?

Author Response

We would like to thank you very much for the opportunity to submit a revision of the aforementioned manuscript for consideration for publication in the Nutrients. We found the comments very helpful and constructive in crafting a revision. We feel confident that the quality of the manuscript has improved as a result of the revision process.

As per your request, we are submitting the document electronically. Below, we respond to each point raised by the reviewer. In the edited manuscript, we have marked all changes using the “Track Changes” function. If we can provide any additional information, or make any additional changes, please do not hesitate to let me know.

Sincerely,

Author Response

(The authors gave the same response as above.)

Reviewer 3 Report

The manuscript of Dr. Enes et al. shows the data of ex-ante risk comparative study to estimate the impact of three tax scenarios, with 100% pass on rate to SSBs on the change in overweight and obesity prevalence.

I have read the manuscript seriously, and I have been interested in the effects of taxation on consumption of SSB and obesity prevalence.

I would like the authors to indicate 3 minor points.

In Abstract, the 2nd and 3rd sentences are "To  estimate  the  impact  of  different  tax  scenarios  for  sugar-sweetened  beverages  (SSBs)  on  the change in overweight and obesity prevalence in the Brazilian population. This is an ex-ante risk comparative study that estimated the impact of three tax scenarios (20, 25 and 30%), with 100% pass on rate to SSBs on the change in overweight and obesity prevalence. ". The meaning of the first sentence was unclear, but it seemed unnecessary. Please correct. And "sugar-sweetened  beverages  (SSBs) " should be put on the 1st line of Abstract.

In manuscript, spaces before unit were sometimes recognaized. For example, page6, paragraph 3.1 " 8kcal for 20% tax to 12kcal", and page 10, Line 39, "25kg/m 2".

Please check and add the space in your manuscript.

Author Response

(The authors gave the same response as above.)

Round 2

Reviewer 1 Report

Corrections make the manuscript clearer to read. Undoubtedly, many more new studies are needed on this particular but important topic.

Reviewer 2 Report

Very good job! Congratulations!